# Progeny Selection to Develop a Sustainable Arabica Coffee Cultivar

**Priscila Carvalho Moreira** [1], **Juliana Costa de Rezende Abrahão** [2,*], **Antonio Carlos da Mota Porto** [3], **Denis Henrique Silva Nadaleti** [2], **Flávia Maria Avelar Gonçalves** [3], **Gladyston Rodrigues Carvalho** [2] **and Cesar Elias Botelho** [2]

1   Department of Agriculture, Federal University of Lavras (UFLA), Lavras 37200-900, MG, Brazil; pricmoreira@yahoo.com.br
2   Southern Minas Regional Unit, Agricultural Research Corporation of Minas Gerais (Epamig), Lavras 37200-900, MG, Brazil; denishenriquesilva@yahoo.com.br (D.H.S.N.); grodriguescarvalho@gmail.com (G.R.C.); cesarbotelho@epamig.br (C.E.B.)
3   Department of Biology, Federal University of Lavras (UFLA), Lavras 37200-900, MG, Brazil; porto.antonio@outlook.com (A.C.d.M.P.); avelar@ufla.br (F.M.A.G.)
*   Correspondence: julianacosta@epamig.br

**Abstract:** The objective of this study was to investigate the most efficient way to select $F_{4:5}$ progenies derived from the Icatu and Catimor groups of *Coffea arabica* and to study the genotypic correlations between the traits related to coffee grain physical quality, diseases, and productivity. A combination of the predicted additive values for production capacity when considering seven harvests and a simultaneous selection for a high sieve percentage and resistance to specific diseases during the last harvest was used. Analyses that selected progenies without considering the genotype × harvest interaction provided inaccurate results, distorting the progeny ranking. Coffee leaf rust and brown eye spot were correlated, showing the possibility of simultaneously selecting for resistance to both diseases. Of the 68 progenies studied here, five showed satisfactory agronomic traits. Our findings will contribute substantially to the development of new coffee cultivars that will allow us to reduce pesticide use.

**Keywords:** *Coffea arabica*; coffee leaf rust; brown eye spot disease

## 1. Introduction

The coffee crop is one of the most important Brazilian commodities, generating income and employment for its many production chain participants, and Brazil is the top coffee exporter in the world. Nevertheless, there are many challenges inherent to farming, such as the risks related to market variation, climate change, plant health maintenance, and plant protection. One of the challenges hindering the expansion of coffee farming is the incidence of diseases and pests. Among them, coffee leaf rust caused by the fungus *Hemileia vastatrix* Berk. et Br [1] and brown eye spot caused by the fungus *Cercospora coffeicola* Berkeley & Cooke [2] both stand out. These pathogens pose relevant phytosanitary problems to coffee plants because they cause serious losses, such as defoliation, reduced productivity and yield, and bean and beverage quality loss [3,4]. It is crucial to control these diseases since their occurrence considerably reduces the profitability of coffee growers.

Sources of resistance identified in coffee species have been used for interspecific hybridization to obtain plants resistant to these pathogens. Germplasms derived from crosses of *Coffea arabica* L. and *Coffea canephora* Pierre ex A. Froehner, such as Icatu and Timor hybrid, are considered sources of genes for resistance to *H. vastatrix* strains [5]. Similarly, Botelho et al. [6] reported the possibility of resistance gene fixation to *C. coffeicola* in the Timor hybrid accessions UFV 377-34 and UFV 376-14 BE 5. However, due to the wide variability in the genome of the fungus *H. vastatrix*, new physiological strains

with virulence genes can emerge that can nullify the resistance of these coffee cultivars, especially in ecosystems highly favorable to the pathogen. Thus, breeders continuously seek to develop new cultivars with durable resistance to the disease.

Progeny tests have been used in genetic parameter estimation and coffee plant selection when evaluating the magnitude and nature of the available variance to quantify and maximize genetic gains. Direct and indirect selection are strategies for obtaining compensating genetic gains, thus providing a basis for selecting superior genotypes and, consequently, shortening the cultivar release time. The efficiency of selection programs that focus on multiple traits depends not only on the type of genetic control of individual traits (i.e., heritability estimates) but also on the genetic correlation between traits [7].

Some studies also emphasize the need to use special methods for predicting genetic values for all candidate individuals as the best strategy for increasing the efficiency of coffee breeding [8,9]. The mixed-model method [10] is a flexible procedure for obtaining estimates of genetic parameters and genetic values, maximizing the correlation between true and predicted genotypic values and genetic gains with selection [11].

The aim of this study was to investigate the most efficient way to select $F_{4:5}$ progenies derived from the Icatu and Catimor groups and to study the genotypic correlations between the traits related to grain physical quality, diseases, and productivity, as knowledge of the magnitude of these correlations can help increase the efficiency of breeding programs. The development of resistant coffee plants can reduce production costs by reducing the number of pesticide applications and improving product quality. However, the success of this strategy depends on the presence of genetic variation in the traits relevant for selection.

## 2. Materials and Methods

Forty-three $F_{3:4}$ progenies derived from crosses between Icatu 3851-2 × Catimor 1509-c8 x, which were obtained through the genetic improvement program at the Agricultural Research Company of Minas Gerais (Empresa de Pesquisa Agropecuária de Minas Gerais-EPAMIG), were evaluated in seven crops in the municipality of São Sebastião do Paraíso, Minas Gerais, Brazil. Based on multiple agronomic traits and resistance to *H. vastatrix* and *C. coffeicola* [12,13], 68 plants from these progenies were selected (Supplementary Materials Table S1), constituting the $F_{4:5}$ generation, and they were planted in two experiments of 34 progenies each along with two commercial cultivars (Catuai Vermelho IAC 99 and Catuai Amarelo IAC 62) as controls. The progenies were divided into two experiments to ensure homogeneous field conditions within each test.

São Sebastião do Paraíso is located in a region with slightly undulating terrain and slopes less than 20% (20°55′ S latitude, 46°55′ W longitude, and 890 m altitude). The soil is classified as a Dystroferric Red Latosol. A 6 × 6 square lattice design was used, with three replicates, 3.2 m spacing between rows, and 0.8 m spacing between plants. Each plot consisted of six plants. The first experiment was arranged with alphanumeric codes from P1 to P34, and the second was arranged from P35 to P68 (Supplementary Materials Table S1). The experiments were set up and conducted according to the agronomic recommendations for coffee crops. Phytosanitary management was conducted effectively and in a timely manner, with the exception of chemical control against coffee leaf rust and brown eye spot, so we could identify and select progenies resistant to *H. vastatrix* and *C. coffeicola*.

In both experiments, grain productivity (PROD) was evaluated for seven consecutive harvests (2012/2013 to 2018/2019). During the last harvest, in addition to the productivity (Hr7), the yield (YI), moka grain percentage (MK), high sieve percentage (HS), and floating grain (FG) percentage, as well as the incidence and severity of coffee leaf rust and brown eye spot were evaluated. The PROD (bags ha$^{-1}$) was calculated by stripping all the cherries, followed by conversion to bags ha$^{-1}$ of processed coffee, according to the actual yield of each genotype, between May and June of each year. For YI, 4 L samples of coffee harvested by total cherry stripping were used, and they were placed in plastic nets and exposed to the sun until reaching approximately 11.0% moisture. The FG percentage was evaluated as proposed by Antunes Filho & Carvalho [14]. For the grain size analysis, a 300 g sample

of processed raw beans was used, which was passed through a set of circular and oblong sieves [15]. The weights of the grains retained in 16, 17, 18, and 19 sieves (HS) and 13, 12, 11, 10, 09, and 08 (MK) were added and converted to percentages.

To determine the incidence and severity of coffee leaf rust and brown eye spot, monthly evaluations were performed between January and June 2018, in which six leaves from the third or fourth pair of plagiotropic branches from five plants were randomly collected, with three on each side of the row and from the middle third of the plants, for a total of 30 leaves per plot. The coffee leaf rust and brown eye spot incidence traits were estimated as the number of leaves with symptoms of coffee leaf rust and/or brown eye spot divided by the total number of leaves in the sample. To assess the severity of coffee leaf rust and brown eye spot, the diagrammatic scales by Capucho et al. [16] and Custodio et al. [17] were used. From the monthly estimated incidence and severity data, the area under the disease progress curve was calculated for coffee leaf rust (CLI), brown eye spot incidence (BEI), coffee leaf rust (CLS), and brown eye spot severity (BES) [18].

*2.1. Statistical Analysis*

The data were analyzed using mixed linear models [10], the variance components were estimated by restricted maximum likelihood, and the genetic values of the progenies were predicted using the best linear unbiased predictor (BLUP). The two experiments were analyzed in a grouped manner, with common controls, during the seventh harvest, using the model $y = Xm + Wb + Bp + Za + e$, where $y$ is the vector of observations; $m$ is the vector of the fixed effects of the experiments; $b$ is the vector of the fixed effects of the replicates; $p$ is the vector of the random effects of blocks within the replicates; $a$ is the vector of the random effects of the progenies and controls; and $e$ is the vector of the random effects of the residuals. Uppercase letters represent the incidence matrices for the aforementioned effects. To analyze the data from all seven harvests, the model $y = Xm + Wb + Bp + Za + Qt + Ul + An + e$ was used, where $t$ is the vector of random effects from the harvest, $l$ is the vector of random effects from the interaction between the progeny/control and harvest, and $n$ is the effect of the random interaction between the harvest and replicate.

When residuals did not meet the assumptions for a Gaussian distribution, response variables were transformed using the Box–Cox procedure. Here, the lambda value estimation ($\lambda$) was based on the likelihood maximization method [19] of the model as adjusted for each variable by treating the factors in the model as fixed:

$$y_t(\lambda) = \begin{cases} \frac{y^\lambda - 1}{\lambda}, & if\ x < 0 \\ \log y, & if\ x \geq 0 \end{cases} \quad (1)$$

where $\lambda$ is the parameter that defines the transformation, $y$ is the variable, and $y\_t$ is the transformed data. For this purpose, we used the Box–Cox function of the MASS package implemented in R [20].

2.1.1. Genetic Parameter Estimates

Based on estimates of the components of variance, the genetic variance between progenies, individual heritabilities, and relative variation coefficient were estimated as described in Resende & Duarte [21]. The broad-sense heritability of average progenies ($h_p^2$) was estimated using the expression $h_p^2 = (1 - (PEV)/\sigma_g^2)$, where (PEV) is the average of error prediction variance of BLUPs and $\sigma_g^2$ is the genetic variance of progenies. The relative coefficient of variation was calculated by identifying the level at which the variability of characters was related to genetic or environmental causes. Likelihood radio tests (LRT) were implemented to verify the significance of random effects [22].

2.1.2. Correlation between Productivity Analysis Strategies

To determine the best way to select for PROD, Spearman's rank correlation between three different approaches was estimated as follows: (i) statistical analyses considering the mean of the seven harvests, (ii) considering only the seventh harvest, and (iii) considering the progeny × harvest interaction.

2.1.3. Correlation between Traits

The genetic correlations between the traits CLI, BEI, CLS, BES, YI, MK, HS, FG, Hr7, and Hr6 (productivity of the sixth harvest) were estimated. The predicted genotypic values of the studied variables were input into a principal component analysis to identify the variables that best explained the variability between the progenies and to quantify the correlations. The predicted genotypic values of the progenies were also used to generate principal component analysis (PCA) biplots in R.

2.1.4. Selection Strategies

The selection gains were estimated based on two selection criteria: direct selection and indirect selection, considering the selection intensity of 20% of top progenies. The expected gains by direct selection for each trait evaluated were estimated by the expression $SG_i = \frac{\sum_{i=1}^{n} BLUP}{n}$, where $i$ is the selected family based on per se performance.

The indirect selection for all traits was calculated using the rank–summation index ($I_{MM}$) proposed by Mulamba and Mock [23], using the equation $I_{(MM)j} = \sum_{i=1}^{n} rij$, where $I_{(MM)j}$ is the value of the index associated with the progeny; $i$ is the ranking of a progeny for the $j$th trait; and $n$ is the number of traits considered in the index. No external economic weight was applied to the traits. The expected gains by indirect selection for each trait evaluated were estimated by the expression $SG_k = \frac{\sum_{k=1}^{n} BLUP}{n}$, where $k$ is the selected family based on the $I_{MM}$ index.

The data were analyzed in R using the lme4 [24], psych [25], and tidyverse [26] packages.

## 3. Results

### 3.1. Genetic Parameter Estimates

After analyzing the assumptions of the linear model [27], we found that the residuals of the CLI, CLS, BEI, and BES traits did not show a normal distribution. These traits were transformed using the Box–Cox procedure, and the analysis was performed again (Table 1). Genetic variation was found for the CLI, CLS, BES, BEI, MK, HS, FG, Hr7, and PROD traits, indicating the possibility of successful selection for these traits ($p > 0.05$). The mean heritability of the progenies was low to moderate for most traits, with the exception of CLI (74%) and HS (71%). The coefficient of relative variation revealed that the selection of the best progenies would allow an increase in the genetic value of the population for the CLI, CLS, BES, MK, HS, and FG traits. No genetic variance was detected for the YI trait, indicating the difficulty of selecting superior genotypes for this trait.

The PROD of the seven consecutive harvests resulted in an overall mean of 34.82 bags ha$^{-1}$, whereas the mean PROD of the seventh harvest was 53.17 bags ha$^{-1}$ (Table 1), suggesting that this was a cycle of high-production capacity progeny, since coffee has biennial production. The same genotypes that stood out in the high-production cycles also stood out in the low-production cycles (Supplementary Materials Figure S1). The mean percentage of FG was 9.7%, that of HS was 26.5%, and that of MK was 23.7% (Table 1).

Figure 1 shows the change in the ranking of the progenies depending on the given analysis. When considering the mean of the seven harvests, the P67, P36, P35, P38, and P61 progenies stood out, with the controls Catuai Amarelo IAC 62 and Catuai Vermelho IAC 99 among the progenies with the worst productive performance. Conversely, when considering only the seventh harvest, the control Catuai Vermelho IAC 99 ranked among the five most productive progenies, along with P49, P23, P63, and P34. When the progeny × harvest

interaction was considered, the P38, P21, P49, P23, and P36 progenies stood out in terms of productivity.

**Table 1.** Estimates of genetic parameters related to the incidence and severity of rust (CLI, CLS) and brown eye spot (BEI, BES), yield (YI), moka grain percentage (MK), high sieve percentage (HS), floating grain (FG) percentage, the productivity of the seventh harvest (Hr7), and the productivity of the seven consecutive harvests (PROD).

| | CLS [1] | BEIn [1] | CLIn [1] | BES [1] | YI |
|---|---|---|---|---|---|
| $\mu$ | 2.55 [a] | 2.68 [a] | 893.62 [a] | 1849.36 [a] | 513.11 |
| $\sigma_g^2$ | 1.41 * | 0.07 * | 58.37 * | 200.18 * | 0.00 [ns] |
| $h_p^2$ | 0.50 | 0.50 | 0.74 | 0.58 | 0.30 |
| $CV_g$ | 342.08 | 14.39 | 39.19 | 39.00 | 0.07 |
| $CV_e$ | 329.29 | 20.28 | 34.83 | 65.18 | 0.00 |
| $CV_r$ | 1.02 | 0.60 | 1.01 | 0.71 | 0.64 |
| | **MK** | **HS** | **FG** | **Hr7** | **PROD** |
| $\mu$ | 23.72 | 26.50 | 9.77 | 53.17 | 34.82 |
| $\sigma_g^2$ | 20.15 ** | 39.62 ** | 19.33 ** | 103.18 * | 14.81 ** |
| $h_p^2$ | 0.61 | 0.71 | 0.61 | 0.37 | 0.38 |
| $CV_g$ | 18.92 | 23.76 | 45.02 | 19.10 | 17.05 |
| $CV_e$ | 25.87 | 25.70 | 62.51 | 41.11 | 51.73 |
| $CV_r$ | 0.73 | 0.92 | 0.72 | 0.46 | 0.21 |

[1] Response variable transformed by $\lambda$ that maximized the log-likelihood ratio [19]; [a] mean calculated with nontransformed values; $\mu$: overall mean; $\sigma_g^2$: genetic variance between progenies; $h_p^2$: mean heritability of the progenies; $CV_g$ coefficient of genetic progenies variation; $CV_e$ coefficient of residual variation; $CV_r$ coefficient of relative variation; *,** and [ns]: significance of the $\chi^2$ test on the likelihood ratio corresponding to the 5% probability level and nonsignificant, respectively.

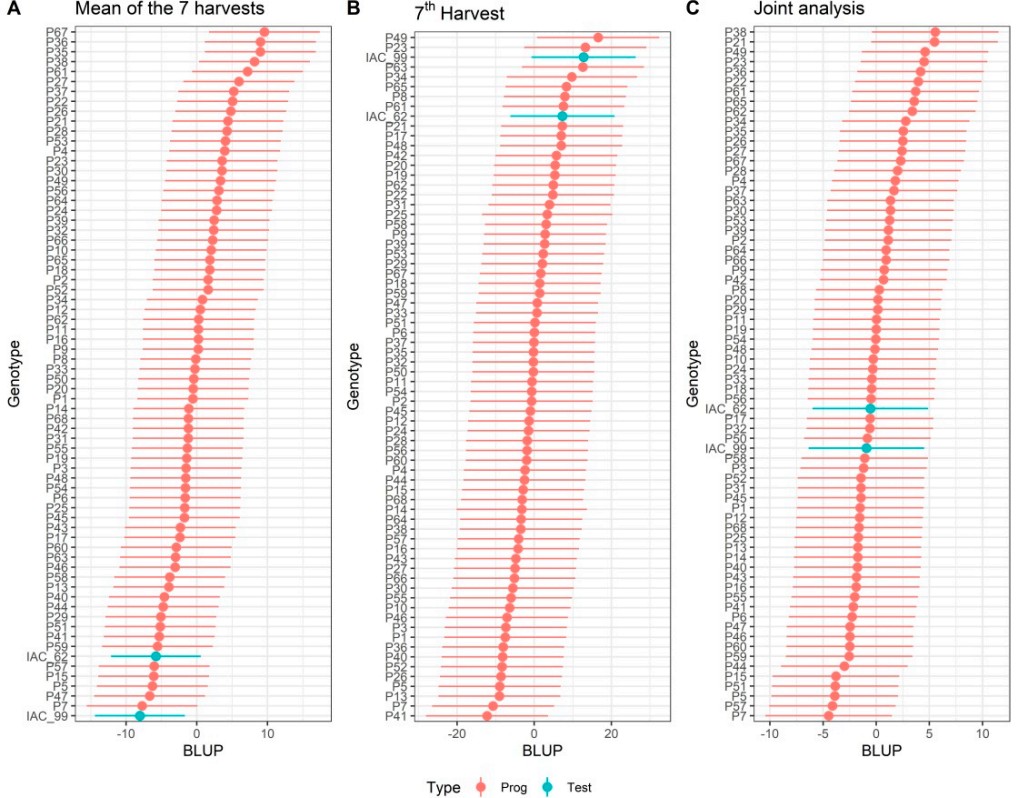

**Figure 1.** Prediction interval for the genetic values (BLUPs) of the $F_{4:5}$ coffee progeny productivity using the following models: (**A**) mean of the seven harvests, (**B**) only the seventh harvest, and (**C**) the harvest × progeny interaction when considering all seven harvests.

### 3.2. Correlation between Productivity Analysis Strategies

To understand the method of analyzing the productivity data and to determine the most efficient way to proceed with the selection, the phenotypic rank correlation (Spearman correlation) was estimated using three different strategies: (A) correlation between the mean of the seven harvests and the harvest × progeny interaction considering all seven harvests, (B) between the seventh harvest only and the harvest × progeny interaction considering all seven harvests, and (C) between the seventh harvest only and the mean of the seven harvests. The correlations between the progeny ranks according to strategies A, B, and C were 0.81, 0.42, and 0.08, respectively (Figure 2).

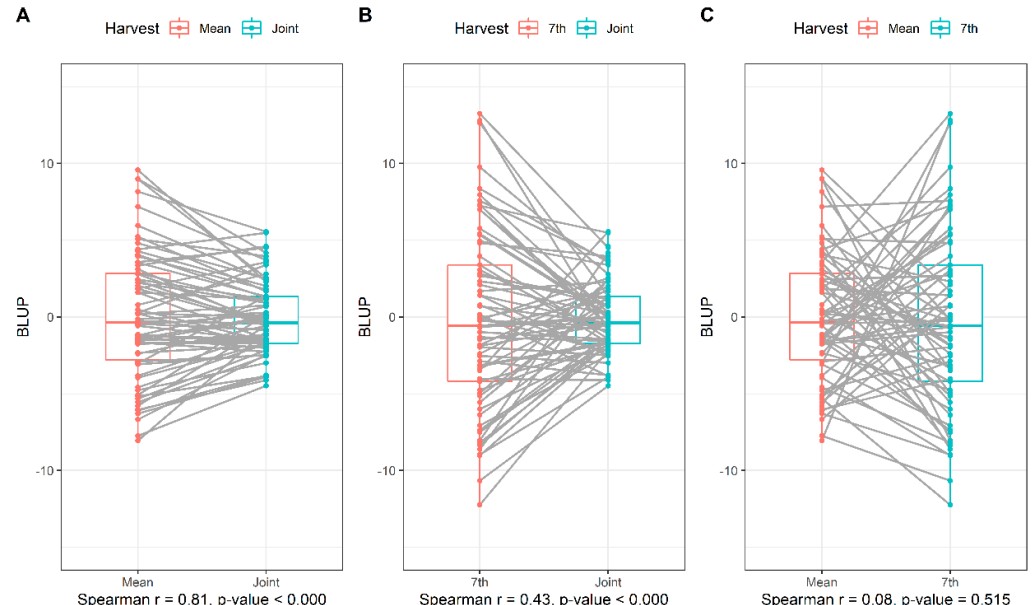

**Figure 2.** Spearman correlation between predicted genetic values (BLUPs) under different strategies for evaluating the productivity of F$_{4:5}$ coffee progenies: (**A**) correlation between the mean of the seven harvests and the harvest × progeny interaction considering all seven harvests, (**B**) between the seventh harvest only and the harvest × progeny interaction considering all seven harvests, and (**C**) between the seventh harvest only and the mean of the seven harvests.

### 3.3. Correlation between Traits

The biplot created from the first two principal components (PC1 and PC2) accounted for approximately 60% of the total variation in the response of the 70 genotypes (68 progenies and two cultivars) among the nine traits evaluated here (Figure 3). PC1 captured 42.6% of the variation, with the productivity and disease response variables contributing more to the variation in this component. The second component (PC2) captured 15.2% of the variation, with a strong contribution from the variation between genotypes for traits related to grain quality (HS and FG).

The P45, P34, and P59 progenies showed high performance in the CLI and CLS traits. P23, P18, and the Catuai Vermelho IAC 99 control were highlighted in HS. P17, P20, and the Catuai Amarelo IAC 62 control were positioned close to BEI and BES. In terms of productivity, the P41, P37, and P64 progenies stood out during the sixth harvest. The P40, P16, and P14 progenies stood out in MK, and the P50, P53, P50, and P44 progenies stood out in regard to the FG percentage, indicating high progeny performance for these traits.

A correlation network graph was plotted (Figure 4). The diseases were positively correlated with the seventh harvest and were negatively correlated with the sixth harvest, and the selection for productivity and disease resistance had no correlated effect on the grain physical quality. Additionally, for each studied disease, the incidence was strongly correlated with the severity, while a weaker correlation was detected between the incidence or severity of one disease and the incidence or severity of the other. The variables related

to grain quality showed weak positive correlations with the disease response variables and weak negative correlations with the productivity responses in Hr6 and Hr7.

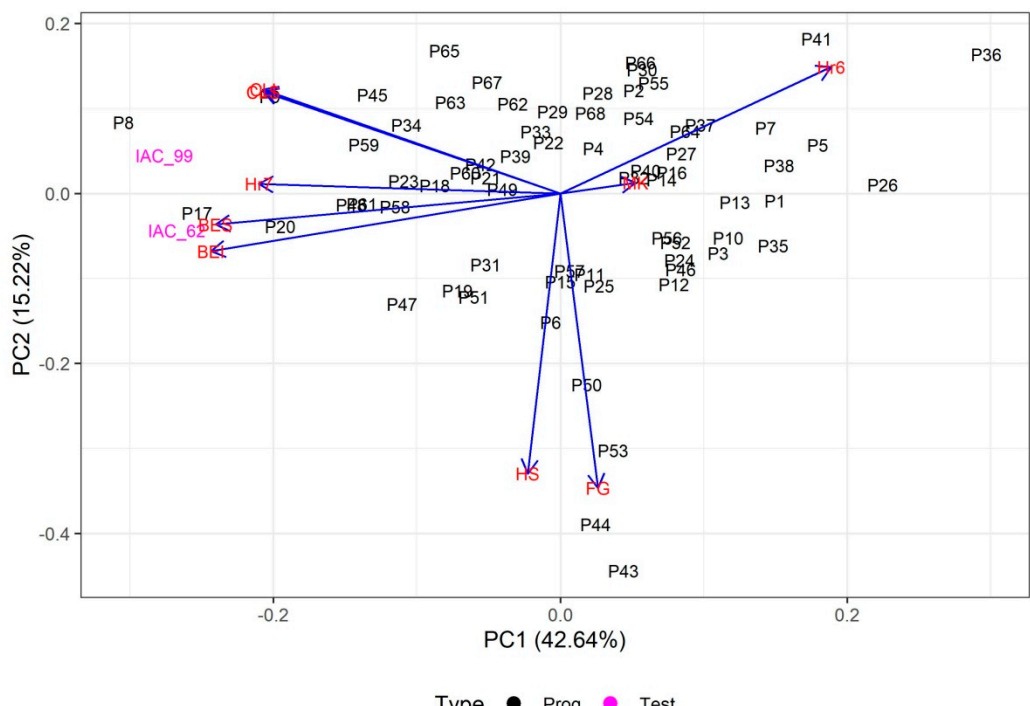

**Figure 3.** Dispersion of the progeny means as a function of the coordinates for the first and second canonical dimensions obtained from nine traits: CLI, CLS, BEI, BES, MK, HS, FG, Hr7, and Hr6.

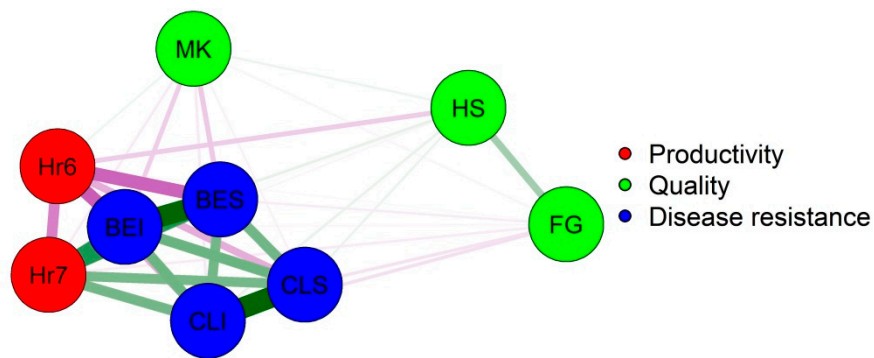

**Figure 4.** Correlation network of traits related to harvest, grain physical quality and diseases, for CLI, CLS, BEI, BES, MK, HS, FG, Hr7, and Hr6. The circles represent the traits, the color of the line indicates positive (green) and negative (red) correlations, and the thickness of the line denotes the magnitude.

*3.4. Selection Strategies*

To select superior progenies and obtain better simultaneous gains for more than one trait of interest, the selection index proposed by Mulamba and Mock [23] was used. The traits evaluated during the seventh harvest were chosen to select the superior $F_{4:5}$ progenies derived from the Icatu × Catimor cross, with the exception of YI, since this trait did not show significant genetic variance. Thus, the sums of "ranks" were used to classify the genotypes in relation to the characters CLI, BEI, CLS, BES, MK, HS, FG, and Hr7. From this classification, the values of each characteristic for the genotypes were added, resulting in a general value considered as $I_{MM}$ (Table 2). To compare the direct and indirect selection gains for the traits, a selection intensity of 20% (14 progenies) was set. When the indices

for each progeny were obtained, P35 [H 29-1-8-5 (III.5) plant 2] had the best performance among the traits of interest, with a mean rank in the eight variables equal to 17.

**Table 2.** Classification of the 14 most promising progenies according to the index proposed by Mulamba and Mock ($I_{MM}$) [23].

| Rank | Progeny | $I_{MM}$ |
|:---:|:---:|:---:|
| 1 | P35 [H 29-1-8-5 (III.5) plant 2] | 17.00 |
| 2 | P26 [H 32-11-17-4 (I.4) plant 2] | 20.37 |
| 3 | P12 [H 32-11-17-4 (III.4) plant 5] | 22.25 |
| 4 | P41 [H 136-1-19-7 (III.28) plant 4] | 22.25 |
| 5 | P1 [H 29-1-9-8 (II.7) plant 1] | 22.75 |
| 6 | P25 [H 32-3-15-20 (I.13) plant 5] | 23.37 |
| 7 | P7 [H 136-1-14-14 (II.23) plant 1] | 23.37 |
| 8 | P36 [H 29-1-8-5 (III.5) plant 3] | 23.62 |
| 9 | P46 [H 136-1-14-10 (I.22) plant 1] | 23.62 |
| 10 | P3 [H 29-1-8-5 (I.5) plant 4] | 23.75 |
| 11 | P44 [H 29-1-8-16 (III.16) plant 3] | 24.12 |
| 12 | P50 [H 32-3-15-20 (II.13) plant 1] | 24.62 |
| 13 | P53 [H 32-3-15-20 (II.13) plant 5] | 26.00 |
| 14 | P10 [H 32-11-17-4 (III.4) plant 1] | 26.62 |

The direct and indirect selection gains based on the use of the sum of ranks (IMM) selection index proposed by Mulamba and Mock [23] were studied (Figure 5). Direct selection had considerable gains in all evaluated traits, ranging from −397.2% for CLI to 27.3% for HS. The results also indicated that FG, MK, BES, BEI, CLS, and Hr7 might have gains in the next generation of 29.7%, −18.0%, −14.0%, −40.0%, 36.8%, and 16.2%, respectively, if the selection were performed in a univariate and direct manner. In contrast, for indirect selection using the IMM, the respective gains in the next generation of 14.0%, 1.8%, −8.0%, −21.7.1%, −29.6%, 312.8%, −9.1%, 6.6%, and 14.7% for FG, MK, BES, BEI, CLS, CLI, Hr7, and HS were calculated.

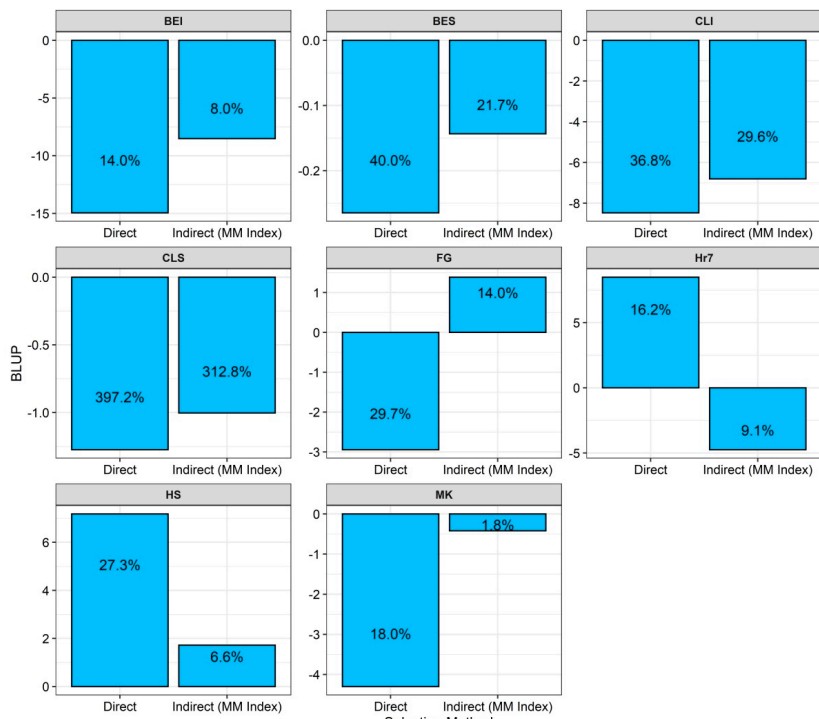

**Figure 5.** Selection gains for the analyzed traits via direct selection and the use of the sum of ranks (IMM) selection index proposed by Mulamba and Mock [23].

## 4. Discussion

Due to the combination of environmental conditions favorable to coffee leaf rust in most coffee farming regions and the wide adoption of susceptible varieties, this disease still represents a significant threat to Brazilian coffee plantations, even 50 years after its introduction to the country. The efforts of several research groups in different regions of the world have been directed toward achieving durable resistance to coffee leaf rust. However, the variability of the pathogen and the emergence of new *H. vastatrix* strains, along with the occurrence of a strain complex, illustrate the evolutionary potential of pathogen populations and their consequent adaptation to the widespread deployment of resistance genes [28].

To address this challenge, improvement programs have combined coffee plants derived from the Icatu and Catimor groups to obtain cultivars with durable resistance to coffee leaf rust since their germplasms have quantitative resistance genes. To achieve this goal, segregating populations obtained by crosses between Icatu and Catimor, which were developed by the Coffee Plant Breeding Program of Minas Gerais, have been grown in the municipalities of Três Pontas [12], Machado [13], São Sebastião do Paraíso, and Campos Altos [29]. In all of these locations, in addition to productivity, the vegetative vigor and the incidence and severity of coffee leaf rust were evaluated, which were used to select the 68 most promising progenies. In the present study, we evaluated these 68 progenies and estimated the genetic parameters for eight traits related to harvest, grain physical quality, and diseases to determine whether there was still any genetic variability within that population to justify continuing the breeding process. After the prediction of the genotypic values of the progenies, for each trait, we investigated the most efficient way to proceed with the selection for productivity and to correlate the eight traits.

When investigating the different ways of analyzing productivity data, the statistical analysis that provided the highest accuracy for progeny selection was the analysis that considered the effect of the progeny × harvest interaction, since this approach took into account not only the highest productivity but also the stability of production [30]. In contrast, the analysis of the performance of the progenies using information only from the last harvest caused bias in the selection, since this performance may be over- or underestimated by the environmental effects under evaluation. This bias could occur because this strategy often inflates the genetic and residual variance of the model, leading to the estimation of genetic parameters more distant from the true ones than other approaches would yield [8,31].

The genetic values of the progenies from the last harvest had a low correlation with the analysis of the harvest × progeny interaction when considering all seven harvests (0.42) as well as with the analysis of the mean of the seven harvests (0.07), demonstrating the low similarity of the strategies in the ranking of the progeny performance what may due the low repeatability (0.21) across the seven harvests. There was a better coincidence of the selected progenies when using the statistical analyses that considered the mean of all seven harvests and the statistical analyses that considered the harvest × progeny interaction (0.81), so this correlation may be good to calculate when data on individual harvests are not available.

In coffee breeding programs, an ideotype is sought for which the performance encompasses, in addition to high production capacity, increased grain size and disease resistance [32], which highlights the need to manage undesirable correlations between traits effectively to maintain genetic variability for future selection. Thus, we studied the genotypic correlation between the CLI, BEI, CLS, BES, MK, HS, FG, Hr7, and Hr6 traits, which yielded genetic parameter estimates significantly different from zero.

The strong positive genetic correlations between the incidence and severity traits suggest collinearity between those components of aggressiveness. Thus, during the exploration many genotypes throughout the breeding process, only the incidence of coffee leaf rust and brown eye spot can be evaluated, making for a quick, easy, more accurate, and more reproducible measure than other quantitative measures, since this approach involves

estimating only the proportion of diseased leaves in a plant; in contrast, "severity" refers to the area of plant tissue affected by diseases based on lesion counts or descriptive scales and is thus more subjective.

The selection process of progenies resistant to diseases may be optimized in this study because there are positive and moderate correlations between coffee leaf rust and brown eye spot. A single small lesion or a few lesions of *C. coffeicola* are enough to cause leaf abscission [33], hindering the work of breeders on cercosporiosis resistance selection. This correlation is not always found, as coffee plants have antagonistic defense mechanisms to neutralize attack by these diseases, in that *C. coffeicola* is a necrotrophic pathogen and *H. vastatrix* is a biotrophic pathogen [34]. Our results suggest that a synergistic interaction between the defense responses to these two organisms may occur.

Arabica coffee takes two years to complete its fruiting phenological cycle, unlike most plants, which complete their reproductive cycle within one phenological year [35]. In coffee plants, grain filling occurs during high-production years at the expense of the formation of new branches, while during low-production years, vegetative branches develop that will produce grains in the following year. Thus, coffee plantations have high and low production in alternate years, a phenomenon called bienniality. This relationship between leaf biomass and coffee productivity is influenced by the occurrence of diseases [36]. In the present study, the sixth harvest, a year of low production (Supplementary Materials Figure S1) was negatively correlated with both diseases. Because it was a year of low production, there was low leaf drop after harvest because productivity is directly related to defoliation levels [37]. With the leafier plants, there was greater shading, which favored a higher incidence of diseases during the next harvest (Hr7). However, the incidence and severity of coffee leaf rust and brown eye spot did not affect productivity because the diseases and Hr7 were positively correlated (Table 1, Supplementary Materials Figure S1).

Regarding the evaluation of coffee crop productivity, a minimum analysis of four consecutive harvests is suggested by several authors to achieve production stability and successful progeny selection [8,9]. For this reason, we analyzed seven harvests to evaluate the productive potential of the progenies and obtain greater efficiency in identifying the most promising ones. According to this evaluation method, which considered the progeny $\times$ harvest interaction effect under a selection intensity of 20%, the 14 selected progenies (P38, P21, P49, P23, P36, P22, P61, P65, P62, P34, P35, P26, P27, and P67) showed grain productivity ranging from 37.12 to 40.38 bags ha$^{-1}$.

These results corroborate other studies that obtained gains in progenies resulting from crosses between *C. arabica* $\times$ *C. canephora* (Icatu, Sarchimor, and Catimor, among others) [38,39], including their reaction to the causal agent of coffee leaf rust (*H. vastatrix*), as many plants are resistant to this fungus. The pyramiding or accumulation of several resistance genes in a single cultivar will make it difficult for the pathogen to break resistance because this would require the loss or masking of its complementary avirulence genes [40]. Despite being very productive, the Catuai Vermelho IAC 99 and Catuai Amarelo IAC 62 controls were not among the selected progenies because these genotypes are susceptible to the studied diseases [6,28].

The analysis allowed us to estimate the mean heritability and to rank the CLI, CLS, BEI, BES, PROD, MK, HS, and FG traits in the seventh harvest for progeny selection. The disease-related traits had high-magnitude heritability and coefficients of relative variation values, indicating reliability in the selection of progenies resistant to coffee leaf rust and brown eye spot.

For all the traits, the percentage gain by indirect selection was lower than that obtained by direct selection. Although direct selection maximizes individual gains, it does not consider gains in other traits [30]. For this reason, for the advancement of generations, the progenies selected by evaluating the harvest $\times$ progeny interaction were considered by accounting for all seven harvests, which coincided with the progenies selected by the IMM (P38, P36, P35, P26, and P27). The match of both selection approaches (multivariate selection and selection by harvest $\times$ progeny interaction) allowed for greater efficiency

in picking out the most promising progenies. These five progenies combined the PROD, resistance to coffee leaf rust and brown eye spot, low FG, and high MK.

In this study, we identified five productive Arabica coffee progenies resistant to coffee leaf rust and brown eye spot and with satisfactory bean size. This selection was performed through the combination of simultaneous selection and selection using predicted additive values by considering the harvest × progeny interaction. Methods that select progenies without considering this interaction may provide inaccurate results, distorting the ranking of progenies. These five selected progenies may contribute substantially to the development of new coffee cultivars with good agronomic traits, which would lead to lower pesticide use.

**Supplementary Materials:** The following supporting information can be downloaded at: https://www.mdpi.com/article/10.3390/agronomy12051144/s1, Figure S1: Correlation network of the seven evaluated harvests; Table S1: $F_{4:5}$ progenies evaluated in the experiments set up at the EPAMIG experimental field in São Sebastião do Paraíso.

**Author Contributions:** Conceptualization, P.C.M. and C.E.B.; writing—original draft preparation, P.C.M., J.C.d.R.A., D.H.S.N., G.R.C., C.E.B., A.C.d.M.P. and F.M.A.G.; writing—review and editing, J.C.d.R.A., A.C.d.M.P. and F.M.A.G. All authors have read and agreed to the published version of the manuscript.

**Funding:** This research was funded by the INCT Café/CNPq, Consorcio Pesquisa Café, and Fapemig.

**Data Availability Statement:** Not applicable.

**Conflicts of Interest:** The authors declare no conflict of interest.

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
