# Peer review of "Progeny Selection to Develop a Sustainable Arabica Coffee Cultivar"

_agronomy, doi:10.3390/agronomy12051144_

Round 1
Reviewer 1 Report
Introduction … adequate
Materials & Methods … As you get into the results section, the authors discuss indexes and statistical packages that are not included in this section (i.e., heritability, CVr, principal components, etc.).
Results …
- The authors need to report only one significant digit when reporting % etc.
- How was heritability defined? Are there standard errors around the heritability estimates that can be reported?
- How is CVr defined?
- Figure 1 is hard to read, and the important points are discussed in the text. I would consider removing this from the manuscript.
- I don’t see the point in Figure 2. The correlations are reported in the manuscript. I recommend removing it.
- Principal Components not mentioned in the M&M section. It would be helpful if the authors reported the eigen vector values (trait weightings).
- On page 7, the authors refer to correlations without reporting the r or p values of those correlations. Figure 4 tells me little regarding the significance and magnitude of the reported correlations.
- Table 2, lists the 14 most promising progenies according to the index … (Mulamba & Mock). They need to define this index. In general, all tables and figures need to stand alone … thus any abbreviation needs to be defined, index in this case needs to be defined.
- Table 2 … direct selection gains or losses of 397.2% seems extremely large … please check the values.
- Again in Figure 5, all the information presented is described within the text. The figure does not add to that discussion. I would delete.
Discussion …
Overall, I found this section very hard to understand and follow.
- lns 319-320 … where these correlations significant?
-
lns 324-325 … ‘… good to calculate when data on individual harvests are not available.’ With a correlation this strong (0.81) why would you use individual data verses mean data? No individual data was presented in the manuscript. Please clarify.
-
ln 332 … ‘… incidence …’ Please describe … it seems like they may be colinear?
-
ln 334 … in lns 245-248 suggests that limited correlation exists between leaf rust and eye spot … why then say that indirect selection for one will improve the other?
-
lns 363-365 … Why did the authors mean of 4 harvests versus 7 harvests …
-
ln 382 … It would be good to report the standard errors around the heritability estimates …
Author Response
- The authors need to report only one significant digit when reporting % etc.
We make this change to the text.
- How was heritability defined? Are there standard errors around the heritability estimates that can be reported?
The Heritability at progeny level was calculated using the following formula: (Resende and Duarte, 2007), where is the average of error prediction variance of BLUPs and is the genetic variance of progenies. We add this information to the text
Still the are no proposal to estimate the stand error of heritability using this way.
- How is CVrdefined?
The relative coefficient of variation is obtained by the ratio between cvg/cve and allows identifying at what level the variability of characters is related to genetic or environmental causes. We add this information to the text
- Figure 1 is hard to read, and the important points are discussed in the text. I would consider removing this from the manuscript.
Figure 1 shows the individual's average one each treatment evaluated on (A) the mean of the seven harvests, (B) only the seventh harvest, and (C) the harvest × progeny interaction. These data do not appear elsewhere in the text. The importance of presenting the individual production data of the studied progenies was highlighted by the second reviewer, therefore, we respectfully request this reviewer to consider the possibility of keeping this figure in the text.
I don’t see the point in Figure 2. The correlations are reported in the manuscript. I recommend removing it.
In addition to correlations, readers can check the gradient of variance of predicted values and the progenies lines of interaction between pair of analysis methods. Thus, we respectfully request this reviewer to consider the possibility of keeping this figure in the text.
- Principal Components not mentioned in the M&M section. It would be helpful if the authors reported the eigen vector values (trait weightings)
We mentioned the PCA analysis in the M&M section. Regarding eigenvector values, we decided not to report them due to the amount of information in the results section, but we added the PCA summary as a supplementary file.
- On page 7, the authors refer to correlations without reporting the r or p values of those correlations. Figure 4 tells me little regarding the significance and magnitude of the reported correlations.
The main goal of the correlation network is not to show the correlations punctually but to show the correlation trend among groups of variables, in this case 'productivity', 'quality' and 'disease resistance'. Anyway, we will make the correlation matrix and p-values available in the supplementary files.
- Table 2, lists the 14 most promising progenies according to the index … (Mulamba & Mock). They need to define this index. In general, all tables and figures need to stand alone … thus any abbreviation needs to be defined, index in this case needs to be defined.
We make this change to the text.
- Table 2 … direct selection gains or losses of 397.2% seems extremely large … please check the values.
We checked twice. The value is correctly.
- Again in Figure 5, all the information presented is described within the text. The figure does not add to that discussion. I would delete.
We understand that this figure draws more attention than the text itself. Reviewer 2 suggested some modifications in the referred figure and all of them were carried ou.t Thus, we respectfully request this reviewer to consider the possibility of keeping this figure in the text.
Discussion …
Overall, I found this section very hard to understand and follow.
- lns 319-320 … where these correlations significant?
We do not use the term ‘significance”, we prefer to use the terms “high” to “low” correlation, since, when comparing the correlation magnitude values of 0.81, 0.42 and 0.07, the superiority of the first value is very evident.
- lns 324-325 … ‘… good to calculate when data on individual harvests are not available.’ With a correlation this strong (0.81) why would you use individual data verses mean data? No individual data was presented in the manuscript. Please clarify.
The preference for the use of individual harvests instead of the average of harvests is highlighted between lines 349 a 357. When considering individual harvests and the interaction between progenies x harvests, greater accuracy is obtained for the selection of progenies, since this approach took into account not only higher yields but also production stability. The values of the individual crops are shown in Figure 1.
- ln 332 … ‘… incidence …’ Please describe … it seems like they may be colinear?
Our results demonstrate a strong genetic correlation between the incidence and severity of both diseases studied. We added the word collinearity to the text as suggested by the reviewer.
- ln 334 … in lns 245-248 suggests that limited correlation exists between leaf rust and eye spot … why then say that indirect selection for one will improve the other?
The suggestion of using the selection of a disease for the benefit of the other is suggested in the specific case of the present work due to the correlation that we found. We evidence this limitation in line 384 and 389.
- lns 363-365 … Why did the authors mean of 4 harvests versus 7 harvests …
In the coffee crop, it is evident that evaluations are necessary in at least four harvests so that we can have reliable results. To increase efficiency, we evaluated seven crops. We rewrote the text to clarify this idea.
- ln 382 … It would be good to report the standard errors around the heritability estimates …
The Heritability at progeny level was calculated using the following formula: (Resende and Duarte, 2007), where is the average of error prediction variance of BLUPs and is the genetic variance of progenies. Still the are no proposal to estimate the stand error of heritability using this way.

Reviewer 2 Report
This manuscript evaluated productivity and disease traits in 68 F4:5 progenies derived from the Icatu and Catimor groups and discussed how to efficiently select F4:5 progenies for data-driven breeding in the coffee crop. Generally, the background information and methods have been clearly described and the results have been clearly conveyed. I recommend that authors have their manuscript checked by an English language native speaker and I have the following comments/suggestions to authors, hopefully, which would be useful to improve the quality of the manuscript.
Lines 35-36: what do Berk. et Br and Berkeley & Cooke mean?
Lines 79-80: What does the “progenies” exactly mean? When you talk about “Forty-three F3:4 progenies” (line 74), do you mean forty-three F3:4 families? Are the 68 plants equivalent to 68 progenies? Why did you split the 68 genotypes into two experiments?
Line 162: “After analyzing the assumptions of the linear model”, this sentence should be improved.
Line 165: the “Genetic variability” seems not an appropriate term.
Line 167 & Line 170: how the mean heritability was calculated should be described in the method section. The genetic variance for YI is zero, why its mean heritability is 0.30?
Line 118: I suggested splitting the “2.1. Statistical analysis” into several subsections each corresponds to one subsection of results and has an independent header, such that readers can easily find how did you get each result when reading the results section.
Line 230: change P453 to P45
Line 272: please describe how the direct and indirect selection gains were calculated in the method section.
Line 301: delete “traits”
Lines 308-325: I agree that selection considering the progeny x harvest interaction is more reliable than selection based on a single harvest. Given this, how does the selection index (Table 2) make sense considering that it was calculated based on the seventh harvest only? How would the breeding decision made from a single harvest affect breeding progress in a real situation? For productivity, since you have data from seven harvests, can you also estimate repeatability?
Lines 341-344: the expression needs to be improved.
Lines 360-362: seem to be contradictory with common knowledge described in lines 36-38.
Lines 387-391: could you show a correlation plot either in the main text or supplemental?
Tables and figures
Table 1: I suggested putting trait names in rows and metrics in columns, and all the variance components used for calculating the mean heritability should be listed.
Figure 2 could you show axes for loadings?
Figure 4: for line colors, there are other colors besides green and red.
Figure 5: If I understood correctly, the height of each bar represents the BLUP and has nothing to do with the selection gain. To avoid confusion, please indicate this in the caption. In addition, it is not necessary to use different colors for different traits since you have a trait name for each bar plot. Did you use the same colors for the studied traits throughout the manuscript?
Author Response
Lines 35-36: what do Berk. et Br and Berkeley & Cooke mean?
Berkeley and Cooke were the responsibles for describle of the fungus Cercospora coffeicola (causal agent of brown eye spot or cercosporiosis)
Berkeley, M. J., & Cooke, M. C. (1876). The fungi of Brazil, including those collected by JWH Trail, Esq., MA, in 1874. Botanical Journal of the Linnean Society, 15(86), 363-398.
Berkeley and Broome were the responsibles for describle of the fungus Hemileia vastatrix (causal agent of coffee rust) in 1869
Berkeley, M.J. and Broome, C.E. (1869) Hemileia vastatrix. Gardeners’ Chronicle, 6, 1157.
It is common to quote the descriptors after inserting the scientific name of pathogens
Lines 79-80: What does the “progenies” exactly mean? When you talk about “Forty-three F3:4 progenies” (line 74), do you mean forty-three F3:4 families? Are the 68 plants equivalent to 68 progenies? Why did you split the 68 genotypes into two experiments?
The term progeny refers to the descendant family of a plant, in the specific case of our work they are endogamous families resulting from self-fertilization. In animal breeding, the term family is used, in plant breeding, the term progeny is used. Among the 43 progenies, we selected 68 plants, which were divided into 2 experiments because they were a large number of plants and we wanted to ensure homogeneous field conditions within each experiment. The common controls allow us these fair comparisons between the two experiments. We add this last information to the text.
Line 162: “After analyzing the assumptions of the linear model”, this sentence should be improved.
To perform an Analysis of Variance there are four basic assumptions: Errors must follow a normal distribution; Errors must be independent; Errors must present constant variance, that is, homogeneity of variances; The model must be additive. Added a reference in the text for clarity.
Line 165: the “Genetic variability” seems not an appropriate term.
The main objective of the work is the selection of progenies. However, the success of this strategy depends on the existence of genetic variation in the traits relevant for selection (lines 59-72). We replaced the term variability with variation
Line 167 & Line 170: how the mean heritability was calculated should be described in the method section. The genetic variance for YI is zero, why its mean heritability is 0.30?
We added the heritability equation in the method session.
Line 118: I suggested splitting the “2.1. Statistical analysis” into several subsections each corresponds to one subsection of results and has an independent header, such that readers can easily find how did you get each result when reading the results section
We make this change to the text.
Line 230: change P453 to P45
We make this change to the text.
Line 272: please describe how the direct and indirect selection gains were calculated in the method section.
We make this change to the text.
Line 301: delete “traits”
We make this change to the text.
Lines 308-325: I agree that selection considering the progeny x harvest interaction is more reliable than selection based on a single harvest. Given this, how does the selection index (Table 2) make sense considering that it was calculated based on the seventh harvest only? How would the breeding decision made from a single harvest affect breeding progress in a real situation? For productivity, since you have data from seven harvests, can you also estimate repeatability?
In this article, there are two ways of selection: selection through interaction and harvests of seven crops and selection (which selected 14 progenies) and selection of mm, (which selected 5 progenies, however considering several traits evaluated by only one year). The five progenies selected by multivariate analysis are included within the 14 selected by interaction, which gives security in the indication of these genotypes. We make this clearer in the text. We got a repeatability (Type-b correlation) of 0.21. We put this information in the text.
Lines 341-344: the expression needs to be improved.
We make this change to the text.
Lines 360-362: seem to be contradictory with common knowledge described in lines 36-38.
Lines 38 to 40 relate the occurrence of the disease and the drop in productivity. Lines 360-362, on the other hand, relate to the increase in productivity and the fall of leaves.
Lines 387-391: could you show a correlation plot either in the main text or supplemental?
These correlations are evidenced in Supplemental Figure S1
Tables and figures
Table 1: I suggested putting trait names in rows and metrics in columns, and all the variance components used for calculating the mean heritability should be listed.
Figure 2 could you show axes for loadings?
We make these change to the text.
Figure 4: for line colors, there are other colors besides green and red.
Actually, is from green (+1) to Magenta (-1). We changed in the text.
Figure 5: If I understood correctly, the height of each bar represents the BLUP and has nothing to do with the selection gain. To avoid confusion, please indicate this in the caption. In addition, it is not necessary to use different colors for different traits since you have a trait name for each bar plot. Did you use the same colors for the studied traits throughout the manuscript?
We make the figure following the recommendations.
